# Mechanical Properties of Multi-Walled Carbon Nanotube/Waterborne Polyurethane Conductive Coatings Prepared by Electrostatic Spraying

**DOI:** 10.3390/polym11040714

**Published:** 2019-04-19

**Authors:** Fangfang Wang, Lajun Feng, Man Lu

**Affiliations:** 1School of Materials Science and Engineering, Xi’an University of Technology, Xi’an 710048, China; wff1170111008@163.com (F.W.); luman569@163.com (M.L.); 2Key Laboratory of Corrosion and Protection of Shaanxi Province, Xi’an 710048, China

**Keywords:** electrostatic spraying, multi-walled carbon nanotubes, waterborne polyurethane coating, dispersity, surface hardness, wear rate, friction coefficient

## Abstract

Electrostatic spraying (ES) was used to prepare multi-walled carbon nanotube (MWCNT)/waterborne polyurethane (WPU) abrasion-proof, conductive coatings to improve the electrical conductivity and mechanical properties of WPU coatings. The dispersity of MWCNTs and the electrical conductivity, surface hardness, and wear resistance of the coating prepared by ES (ESC) were investigated. The ESC was further compared with coatings prepared by brushing (BrC). The results provide a theoretical basis for the preparation and application of conductive WPU coatings with excellent wear resistance. The dispersity of MWCNTs and the surface hardness and wear resistance of ESC were obviously better than those of BrC. With an increase in the MWCNT content, the surface hardness of both ESC and BrC went up. As the MWCNT content increased, the wear resistance of ESC first increased and then decreased, while the wear resistance of BrC decreased. It was evident that ESC with 0.3 wt% MWCNT was fully capable of conducting electricity, but BrC with 0.3 wt% MWCNT failed to conduct electricity. The best wear resistance was achieved for ESC with 0.3 wt% MWCNT. Its wear rate (1.18 × 10^−10^ cm^3^/mm N) and friction coefficient (0.28) were the lowest, which were 50.21% and 20.00% lower, respectively, than those of pure WPU ESC.

## 1. Introduction

Waterborne polyurethane (WPU) with water as the dispersion medium is a class of eco-friendly coatings [1,2,3]. It does not volatilize organic solvents into the air and is now widely being used in the industry to gradually replace solvent-based polyurethane owing to its environmentally friendly characteristic [4,5,6,7]. However, the poor mechanical strength and performance of WPU and its inability to conduct electricity may restrict its applications in some working conditions where relatively high antistatic property and wear resistance is required. Therefore, it is necessary for WPU to be modified to meet the requirements of harsh conditions [8]. Researchers have previously shown that adding nanoparticles to polymers to prepare organic/inorganic nanocomposites could strengthen the physical and mechanical properties of polymers [9,10]. As a kind of carbon materials, multi-walled carbon nanotubes (MWCNTs) holding the performance of conducting electricity are likely to maintain resistance to generally chemical corrosive media. The addition of MWCNTs in WPU could therefore effectively improve the electrical conductivity and other mechanical properties of WPU coatings. Moreover, some polar groups, such as –OH, may be adsorbed on the surface of the MWCNT due to the fibrous structure of the material and its outstanding surface activity. The crosslinking reaction that occurs between these polar groups and some polar groups in the molecular chain of WPU during the curing process of the coating could make the WPU coating form a crosslinked network structure [11], thus enhancing the mechanical property of the composite coating [12]. Khun et al. [13] prepared PU composite coatings with different MWCNT contents and found that the cathodic delamination of PU coatings was significantly lessened as the MWCNT content increased to 0.5 wt%. Manas-Zloczower et al. [14] obtained PU nanocomposites with the addition of MWCNTs via the in-situ polymerization of 1,4-phenyldiisocyanate (PPDI) and polycaprolactonediol, it was concluded that the dispersity of nano-fillers was well and properties of obtained nanocomposites were superior. Gao et al. [15] prepared a flexible conductive polymer nanofiber composite (FCPNC) with the addition of carbon nanotube (CNT) and found that the good electrical conductivity and interconnected porous structure of the FCPNC made it possible to be used as a chemical vapor sensor. However, MWCNTs are inclined to aggregate due to their characteristics of high aspect ratio and specific surface area [16]. When the MWCNT content is low, the number of agglomerated MWCNTs might be reduced to a certain extent. However, the electrical conductivity of the composite coating may be too poor to meet the requirement of antistatic performance, and its surface hardness and wear resistance may also be weak. When the MWCNT content is high, the antistatic property of the coating can satisfy the application requirements [17,18,19], but the composite coating structure would be loose, and the bond strength of the coating to the metal substrate may be poor. Thus, the coating will likely peel off in pieces once it is subjected to external forces during application.

In electrostatic spraying (ES), there is a nozzle with a spiculated edge in the head of the spray gun, which instantly generates high-voltage discharge as well as air ionization once the high-voltage negative electricity is connected. The advantage of high-voltage corona discharge is that it leads to the formation of electrostatic field between the spray gun and the metal substrate. Consequently, the liquid coating with negative charges ejected from the nozzle is attracted to the metal substrate with positive charges under the action of electrostatic attraction. If compressed air would be used as the driving force to transport the liquid coating, the coating could be more effectively atomized with the impact force of compressed air during spraying, it may prevent the agglomeration of conductive fillers to some extent. In addition, ES could overcome the problem of controlling the coating thickness, which is difficult for brushing (Br). In this work, a series of MWCNT/WPU nanocomposite coatings were prepared by ES to not only enhance the dispersity of conductive fillers to some extent but also promote the antistatic and mechanical properties of the WPU coating. The dispersity of MWCNTs and the electrical conductivity, surface hardness, and wear resistance of the coating were studied. The result was further investigated by comparing the coatings with those prepared by Br. This work provides a theoretical basis for the preparation and application [20,21] of MWCNT/WPU abrasion-proof, conductive coatings.

## 2. Materials and Methods

### 2.1. Experimental Materials

WPU was supplied by Jining Huakai Resin Co. Ltd., Jining, China. Its volatile organic compound (VOC) concentration, viscosity, and solid content were 253 g/L, 75 cps, and 35%, respectively. The MWCNTs (FloTube 9000 series) were supplied by Beijing Tiannai Technology Co., Ltd., Beijing, China. The purity, average diameter, average length, and tap density of MWCNTs were 95–97.5%, 10–15 nm, 10 μm, and 0.03–0.15 g/cm^3^, respectively.

MWCNTs with different contents (0, 0.3, and 0.6 wt%) were each added in WPU by an 85-2 magnetic stirring device (Hangzhou Instrument Motor Co., Ltd., Hangzhou, China) at a low speed of 200–300 r/min for 30 min. Then, the mixtures were each treated using a KQ-50B ultrasonic dispersion device (Kunshan Ultrasonic Instrument Co., Ltd., Kunshan, China) for 30 min. The relatively evenly treated WPU dispersions with different MWCNT contents were obtained.

Q235 steel with a size of 50 × 20 × 3 mm was used as the metal substrate and was roughened by a YX-6050A sand blasting device (Anbangruiyuxin Machine Technology Development Co. Ltd., Wuhan, China). The process condition of sand blasting treatment was as follows. The air pressure was controlled at 0.6–0.8 MPa, the distance between the spray gun and the metal substrate was kept at 110–150 mm, and the time of sand blasting treatment was kept at 30–40 s.

### 2.2. Preparation of Coatings

#### 2.2.1. Coating Prepared by ES (ESC)

Due to the characteristic of self-adjustment of coating thickness, the thickness of ES cannot be increased any further after reaching a certain thickness. Therefore, a method of multi-spraying was adopted to obtain a thicker coating. The obtained MWCNT/WPU dispersions were each sprayed on roughened metal substrates to form an underlayer coating using a NEW KCI-CU801 electrostatic spraying equipment (Shenzhen Honghaida Instrument Co., Ltd., Shenzhen, China). As the underlayer coating was in semidry and nonflowing conditions, the same MWCNT/WPU dispersion as the underlayer coating was sprayed again on the uncured underlayer coating to prepare an upper-layer coating. The average thickness of the multilayer coating was controlled at 80–87 μm. The samples (the Q235 steel substrate with a multilayer coating) were first cured at room temperature for 3 days and then at 70 °C for 24 h in an oven (Zhejiang YuyaoYuandong CNC Instrument Factory, Yuyao, China). The process condition of ES was as follows. The voltage of ES was set at 50–60 KV, the pressure of the compressed air was kept at 0.6–0.7 MPa, the distance between the spray gun and the Q235 steel substrate was controlled at 100–120 mm, the feedwell diameter was 1 mm, the liquid flow rate was 2 mL/min, and the spray time was 1–2 min.

#### 2.2.2. Coating Prepared by Br (BrC)

The obtained WPU dispersions with different MWCNT contents were each brushed on roughened metal substrates to prepare MWCNT/WPU composite coatings. The average thickness of the coating was 80–87 μm. The samples were first cured at room temperature for 3 days and then at 70 °C for 24 h in an oven.

### 2.3. Measurements

#### 2.3.1. Wear Resistance of the Coating

The wear resistance of the coating, which was tested according to ASTM G99-05, was evaluated by its wear rate and friction coefficient [22,23,24]. The experiment on the wear was conducted with an HT-1000 high-temperature scratch testing machine (Lanzhou Zhongke Kaihua Development Co., Ltd., Lanzhou, China) at room temperature using the Q235 steel substrate (50 × 20 × 3 mm) with the coating against a steel bearing ball (Φ2.5 mm) with a hardness level of HRC62. The applied load was 4 N, the rotation speed of the steel ball was 400 r/min, the sliding radius was 7 mm, and the wear time was 10 min. The wear rate was evaluated using Equation (1):*I* = *Δm*/2*πrntFρ*(1)
where *I* is the specific wear rate (cm^3^/mm N), *Δm* is the loss weight (g), *r* is the sliding radius (mm), *n* is the rotation speed of the steel ball (r/min), *t* is the wear time (min), *F* is the applied load (N), and *ρ* is the density of the WPU coating (g/cm^3^).

#### 2.3.2. Surface Hardness of the Coating

The surface hardness of the coating was measured using a LX-A Shore durometer with a measurement range of 0–100 HA (Leqing Sanwen Metering and Detection Device, Wenzhou, China). The average value was calculated by five data points.

#### 2.3.3. Electrical Conductivity of the Coating

The electrical conductivity of the coating was evaluated by its resistivity, which is equivalent to the multiplication of the thickness and the square resistance. The coating thickness was tested using a HCC-18 magnetoresistive thickness meter (Shanghai Huayang Testing Instrument Co., Ltd., Shanghai, China). The square resistance was tested at room temperature using a DY2101 digital multimeter (Duoyi Multimeter, Xi’an, China). The average value of each parameter was counted by six data points.

#### 2.3.4. Micromorphology of the Coating

The morphology of the MWCNT in the uncured MWCNT/WPU coating that was just brushed or sprayed on the surface of the steel substrate was observed using a JEM-3010 high-resolution transmission electron microscope (TEM) (JEOL, Tokyo, Japan) to characterize the dispersion of MWCNTs in the WPU resins. The cross-sectional morphology of the coating was investigated using a Merlin Compact scanning electron microscope (SEM) (Zeiss, Oberkochen, Germany) to characterize the dispersion of MWCNTs in the coating.

When the test on the wear was finished, the surface morphology of the wear track was observed with a VEGA3 XMU SEM (TESCANSCAN, Brno, Czech) to characterize the effect of ES on the wear resistance of the water-based conductive coating.

## 3. Results and Discussion

### 3.1. Dispersity of MWCNTs

Figure 1 shows cross-sectional morphologies of WPU coatings with different MWCNT contents. When the MWCNT content was 0.3 wt% (Figure 1A), the MWCNTs in ESC were relatively evenly dispersed without obvious agglomeration and sedimentation. Although MWCNTs did not come into contact with each other, the generation of tunneling effect from the close average distance between them would enable the composite coating to conduct electricity [25]. As the MWCNT content increased to 0.6 wt% (Figure 1B), there were no apparently agglomerated MWCNTs in the ESC, and they intertwined with each other to form an infinite conductive network. The presence may be explained as follows. Due to the good atomization performance of ES, the still-agglomerated MWCNTs that had been treated by magnetic stirring and ultrasonic dispersion could be dispersed again. Thus, the agglomeration and sedimentation of MWCNTs could be weakened to some extent. Moreover, it was obvious that the morphology of BrC was different from that of ESC. When the MWCNT content was 0.3 wt%, there were few MWCNTs in WPU resins in the top area of BrC (Figure 1C), resulting in BrC being unable to conduct electricity. As the MWCNT content increased to 0.6 wt%, MWCNTs in BrC came into contact with each other to form a valid conductive network (Figure 1D). However, the MWCNTs were apparently agglomerated and deposited due to their uneven distributions, and the structure of BrC was thus less compact. The dispersion of MWCNTs in WPU resins was further studied by TEM. Figure 2A,B show the morphologies of MWCNTs in uncured ESC with 0.6 wt% MWCNT and BrC with 0.6 wt% MWCNT, respectively, on the surfaces of the steel substrates. Compared with the morphology of the MWCNT in ESC with 0.6 wt% MWCNT (Figure 2A), there were obvious agglomerated MWCNTs in BrC with 0.6 wt% MWCNT (Figure 2B), and the size of the agglomerated MWCNT particles was about 150–200 nm. The result was consistent with that of SEM.

Table 1 summarizes the electrical conductivity of the coatings. It is evident that ESC with 0.3 wt% MWCNT was fully capable of conducting electricity, but BrC with 0.3 wt% MWCNT failed to conduct electricity. The electrical conductivity of ESC was better than that of BrC. Generally, the electrical conductivity results of the coatings were consistent with that of the dispersity of MWCNTs.

### 3.2. Surface Hardness

The surface hardness of the WPU coating with different MWCNT content is shown in Figure 3. It is evident that the surface hardness of ESC was significantly higher than that of BrC with the same MWCNT content. As the MWCNT content rose, the surface hardness of both ESC and BrC went up. The difference was that the growth rate of the surface hardness of ESC was more rapid than that of BrC. The surface hardness of pure WPU ESC (79.5 HA) increased by 8.9% compared with that of pure WPU BrC (73.0 HA). The reason may be explained as follows. Generally, the reason the structure of ESC is more compact is that it is under the action of high-voltage electrostatic field. The defects that are generated in the coating during the former spray can be filled up by the atomized coating during the latter spray. However, when compressed air is used as the driving force to transport the liquid coating, the liquid coating may be attracted to the metal substrate by electrostatic attraction during spray, and it is also more effectively atomized by the impact force of compressed air, thus preventing agglomeration of conductive fillers. A denser coating structure is therefore formed after curing. Moreover, some of the moisture in the WPU coating is taken away during the spraying process, and the micropores caused by moisture volatilization during the curing process of WPU thus falls. Therefore, the coating structure becomes denser, and the surface hardness of the coating is strengthened. In our study, as the MWCNT content increased from 0.3 wt% to 0.6 wt%, the surface hardness of ESC increased by 6.6%, but the surface hardness of BrC merely increased by 4.0%. The reason the growth rate of the surface hardness of ESC was higher than that of BrC may be because the MWCNTs in the coating were relatively evenly dispersed with the help of ES, which effectively prevented the MWCNTs from depositing and agglomerating [26]. The MWCNTs with hardness greater than that of WPU resins filled up the micropores arising from the curing process of WPU, and the hardness of the WPU composite coating was thus greatly improved. In contrast, the ununiform dispersion of MWCNTs in BrC resulted in MWCNTs being more likely to agglomerate and deposit, and there might have been few or no MWCNTs in the WPU resins in the upper area of the coating. This intensified the formation of defects in the coating and thus weakened the effect additional MWCNTs might have had on improving the surface hardness of the coating.

### 3.3. Wear Resistance

Figure 4 and Figure 5 show the wear rates and friction coefficient–time curves of WPU coatings with different MWCNT contents. During the wear test, the applied load was 4 N, the rotation speed of the steel ball was 400 r/min, the sliding radius was 7 mm, and the wear time was 10 min. Figure 6 shows the wear track morphologies of WPU coatings with different MWCNT contents. It can be concluded from Figure 4 and Figure 5 that, as the MWCNT content increased, the wear rate and friction coefficient of ESC first decreased and then increased, meaning that its wear resistance first increased and then decreased. However, the wear rate and friction coefficient of BrC accordingly increased, which meant that its wear resistance declined. It is apparent that the wear resistance of ESC was lower than that of pure WPU BrC when the MWCNT content was less than 0.6 wt%. The wear rate (2.37 × 10^−10^ cm^3^/mm N) and friction coefficient (0.35) of pure WPU ESC decreased by 50.00% and 10.26%, respectively, compared with those of pure WPU BrC.

By analyzing the wear rate and friction coefficient, it was apparent that the best wear resistance was ESC with 0.3 wt% MWCNT (Figure 4 and Figure 5). Its wear rate (1.18 × 10^−10^ cm^3^/mm N) and friction coefficient (0.28), which were the lowest among all coatings, reduced by 50.21% and 20.00%, respectively, compared with those of pure WPU ESC. The main reason for this is that, when the MWCNT content was 0.3 wt%, the relatively evenly dispersed MWCNTs in WPU resins would have compensated micropores in ESC, thus enhancing the strength of the coating [27]. Therefore, it would be impossible to peel off the ESC from the steel substrate in pieces during the wear process, and its friction coefficient–time curve was relatively smooth. When the MWCNT content was lower than 0.3 wt%, the strength of the coating decreased, and the coating would be more inclined to wear. As the MWCNT content increased to 0.6 wt%, the surface hardness of the composite coating was greater than that of ESC with 0.3 wt% MWCNT owing to the greater hardness of the MWCNT. However, the large addition of MWCNTs would cause a part of the MWCNT to agglomerate and the number of worn pieces may go up during the wear test. The formation of worn pieces may not only greatly aggravate the wear rate of the coating but also come into the wear track to become a sort of abrasive material that exacerbated the wear of the coating and generated a wavy friction coefficient–time curve. Figure 6A–C show wear track morphologies of ESC after the wear test, which may further explain the results of the wear experiment. It is obvious that there were fewer worn pieces on the wear track of pure WPU ESC. Its wear track seemed to be shallow, and there were almost no block pieces on the wear track of ESC with 0.3 wt% MWCNT. However, there were a large number of flake-like pieces on the wear track of ESC with 0.6 wt% MWCNT.

The analysis of the wear rate and friction coefficient curves of BrC (Figure 4 and Figure 5) showed that the wear resistance of BrC was worse than that of ESC, and the friction coefficient–time curve of BrC was very wavy. This may be due to the uneven dispersion of MWCNTs and the poor compactness of BrC. In addition, the upward trend of agglomerated MWCNTs with the increase in the MWCNT content may have increased the microdefects in BrC. As the friction time increased, the number of worn pieces in the coating also increased. It came into the wear track as a kind of abrasive material, thus increasing the wear rate and friction coefficient of BrC. Figure 6D–F show wear track morphologies of BrC after the wear test. It is evident that the wear track of BrC was wider and deeper than that of ESC, and the number of worn pieces on the wear track of BrC was greater compared with that of ESC. Additionally, the degree of wear of BrC was aggravated with the increase in the MWCNT content.

## 4. Conclusions

It can be concluded that the dispersity of MWCNTs and the surface hardness and wear resistance of ESC were obviously better than those of BrC. When the MWCNT content increased, the surface hardness of both ESC and BrC went up. The wear resistance of ESC first increased and then decreased, while the wear resistance of BrC decreased. When the MWCNT content was only 0.3 wt%, the coating prepared by ES was capable of conducting electricity, but the coating prepared by Br failed to conduct electricity. The best wear resistance was achieved for ESC with 0.3 wt% MWCNT. Its wear rate (1.18 × 10^−10^ cm^3^/mm N) and friction coefficient (0.28) were the lowest, which were 50.21% and 20.00% lower, respectively, than those of pure WPU ESC.

## Figures and Tables

**Figure 1 polymers-11-00714-f001:**
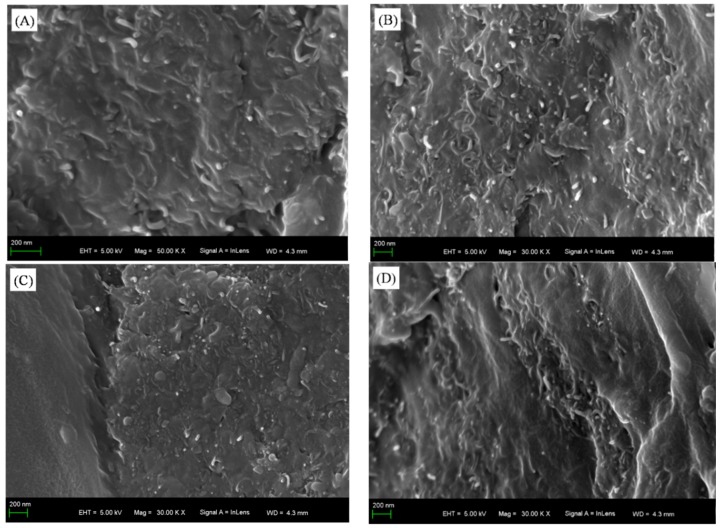
Cross-sectional morphologies of waterborne polyurethane (WPU) coatings with different multi-walled carbon nanotube (MWCNT) contents: (**A**) coating prepared by electrostatic spraying (ESC) with 0.3 wt% MWCNT, (**B**) ESC with 0.6 wt% MWCNT, (**C**) coating prepared by brushing (BrC) with 0.3 wt% MWCNT, and (**D**) BrC with 0.6 wt% MWCNT.

**Figure 2 polymers-11-00714-f002:**
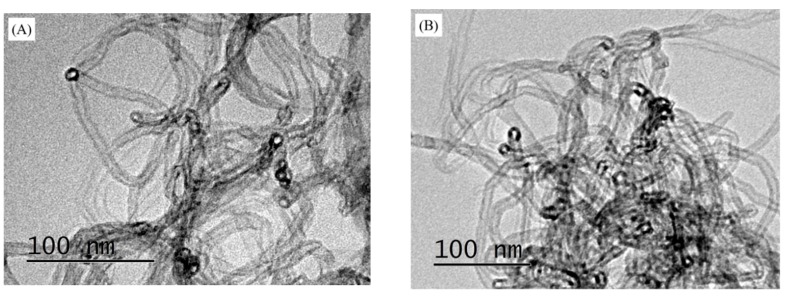
Morphologies of MWCNTs in uncured (**A**) ESC with 0.6 wt% MWCNT and (**B**) BrC with 0.6 wt% MWCNT on the steel substrates.

**Figure 3 polymers-11-00714-f003:**
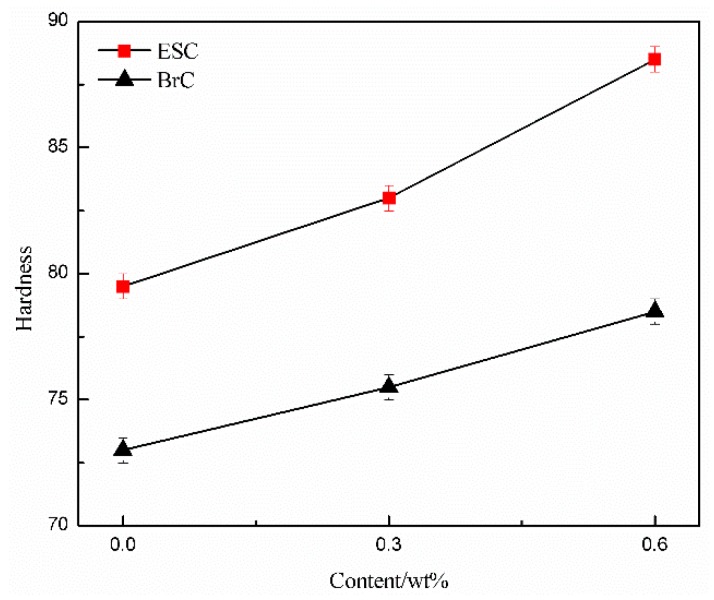
Surface hardness of ESC and BrC with different MWCNT content.

**Figure 4 polymers-11-00714-f004:**
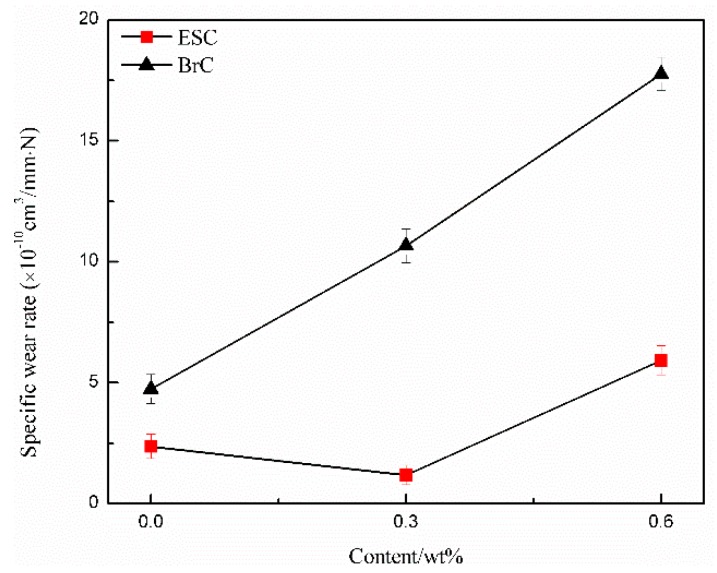
Wear rate curves of ESC and BrC with different MWCNT content.

**Figure 5 polymers-11-00714-f005:**
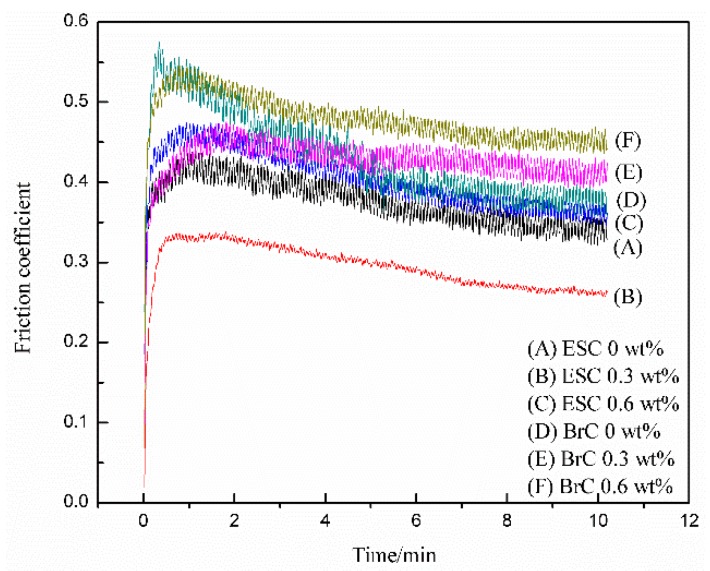
Friction coefficient—time curves of WPU coatings with different MWCNT contents.

**Figure 6 polymers-11-00714-f006:**
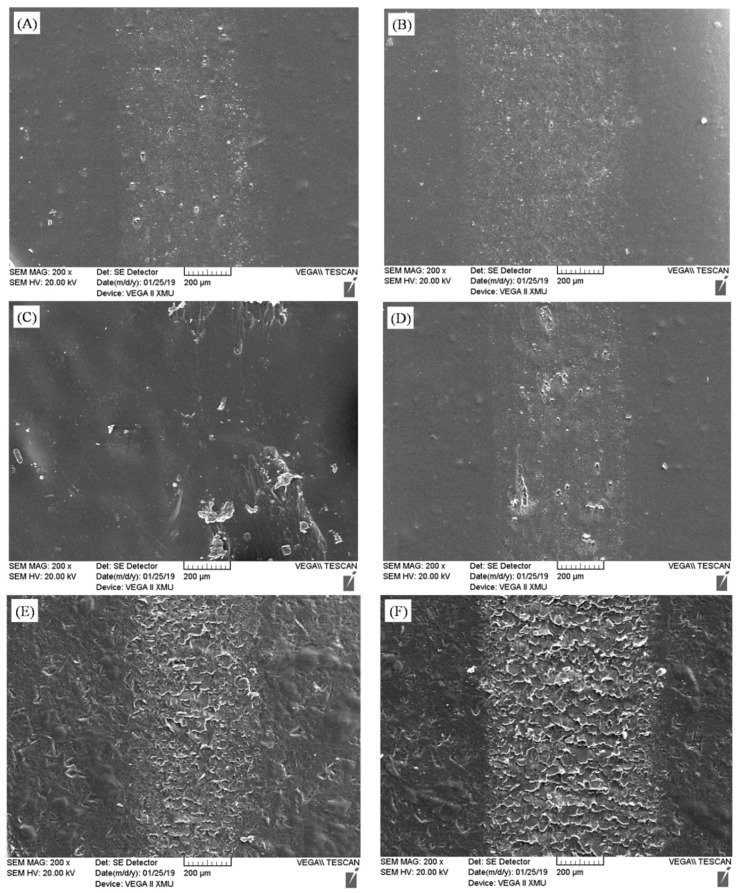
Wear track morphologies of coatings with different MWCNT contents: (**A**) ESC with 0 wt% MWCNT, (**B**) ESC with 0.3 wt% MWCNT, (**C**) ESC with 0.6 wt% MWCNT, (**D**) BrC with 0 wt% MWCNT, (**E**) BrC with 0.3 wt% MWCNT, and (**F**) BrC with 0.6 wt% MWCNT.

**Table 1 polymers-11-00714-t001:** Electrical conductivity of WPU coatings with different MWCNT contents.

Coatings	Properties	0 wt%	0.3 wt%	0.6 wt%
ESC	Thickness (μm)	81 ± 4	82 ± 4	83 ± 4
	Square resistance (MΩ)	0	156.2 ± 5	2.6 ± 0.2
	Resistivity (Ω m)	0	12,808.4	215.8
BrC	Thickness (μm)	82 ± 4	82 ± 4	83 ± 4
	Square resistance (MΩ)	0	0	155.7 ± 0.5
	Resistivity (Ω m)	0	0	12,923.1

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
