# Peer review of "Mechanical Properties of Multi-Walled Carbon Nanotube/Waterborne Polyurethane Conductive Coatings Prepared by Electrostatic Spraying"

_polymers, 2019, doi:10.3390/polym11040714_

Round 1
Reviewer 1 Report
Dear authors,
The presented manuscript describes the preparation of waterborne polyurethanes modified with MWCNTs. The concept is sound and despite the fact that the field has extensively studied in literature the comparison of different techniques to apply the coatings over steel substrates might be of interest for potential readers of Polymers.
However, the manuscript lacks of some relevant data and discussion to be considered for publication in its current form.
Dear authors,
The presented manuscript describes the preparation of waterborne polyurethanes modified with MWCNTs. The concept is sound and despite the fact that the field has extensively studied in literature the comparison of different techniques to apply the coatings over steel substrates might be of interest for potential readers of Polymers.
However, the manuscript lacks of some relevant data and discussion to be considered for publication in its current form.
GENERAL REMARKS:
The introduction needs a more detailed revision of the current state of the art of the use of MWCNTs on the modification of polyurethanes which has been extensively studied. The use of electrospraying might be interesting for the application of these coatings, although it is not sufficiently justified in the manuscript. My recommendation is to revise and re-write the introduction considering all these points.
SPECIFIC REMARKS:
- Please provide information about the production of the MWCNTs, do they content metals? Which quantities, surface area, etc….
- Describe the nozzle of the spray gun. Do you appreciate agglomeration on the tip?
- The discussion of the dispersion of the MWCNTs by SEM is not sufficiently justified. Please provide information of aggregate sizes, etc… micrographs c and d are clearly inhomogeneous which may lead to not conclusive results.
- Please provide the electrical conductivity of the coatings, no values are given in the manuscript which are necessary to discuss their properties.
My recommendation to
the editor will be to reject the manuscript in its current form and to
revise it after including all those required modifications.
With kind regards
GENERAL REMARKS:
The introduction needs a more detailed revision of the current state of the art of the use of MWCNTs on the modification of polyurethanes which has been extensively studied. The use of electrospraying might be interesting for the application of these coatings, although it is not sufficiently justified in the manuscript. My recommendation is to revise and re-write the introduction considering all these points.
SPECIFIC REMARKS:
- Please provide information about the production of the MWCNTs, do they content metals? Which quantities, surface area, etc….
- Describe the nozzle of the spray gun. Do you appreciate agglomeration on the tip?
- The discussion of the dispersion of the MWCNTs by SEM is not sufficiently justified. Please provide information of aggregate sizes, etc… micrographs c and d are clearly inhomogeneous which may lead to not conclusive results.
- Please provide the electrical conductivity of the coatings, no values are given in the manuscript which are necessary to discuss their properties.
My recommendation to the editor will be to reject the manuscript in its current form and to revise it after including all those required modifications.
With kind regards
Author Response
Response to Reviewer 1 Comments
Dear editor and reviewer,
Thank you for your comments concerning our manuscript entitled " Mechanical properties of multi-walled carbon nanotubes/waterborne polyurethane conductive coatings prepared by electrostatic spraying " (Manuscript ID: polymers-481645). Thanks for the positive, helpful and constructive comments and suggestions. We elaborately revised the manuscript according to these comments, where the revised parts have been highlighted (in red) for the convenience of checking. The following responses are in accordance with the specific comments in the report.
If you have any questions regarding this revision, please contact me.
Sincerely yours,
Lajun Feng
A point-by-point reply to the reviewer's comments:
Point 1: English language and style: English language and style are fine/minor spell check required.
Response 1: Thanks for the helpful and kind suggestion. The authors have checked the manuscript carefully. In the revised manuscript, the English in the whole article has been modified and some grammatical mistakes have also been corrected.
Point 2: The introduction needs a more detailed revision of the current state of the art of the use of MWCNTs on the modification of polyurethanes which has been extensively studied. The use of electrospraying might be interesting for the application of these coatings, although it is not sufficiently justified in the manuscript. My recommendation is to revise and re-write the introduction considering all these points.
Response 2: Thanks for the thoughtful suggestion. The introduction has been re-written according to the Reviewer’s suggestion. A more detailed revision of the current state of the art of the use of MWCNTs on the modification of polyurethanes which has been extensively studied has been improved in the revised manuscript as"The addition of MWCNTs in WPU could effectively improve the electrical conductivity and other mechanical properties of the WPU coating. Moreover, some polar groups such as -OH may be adsorbed on the surface of the MWCNT due to the fibrous structure of the MWCNT and its outstanding surface activity, then, the crosslinking reaction that occurred between these polar groups with some polar groups in the molecular chain of WPU during the curing process of the coating could make the WPU coating form a crosslinked network structure [11] and, thus, enhance the mechanical property of the composite coating [12]. Khun, et, al [13] prepared PU composite coatings with different MWCNT contents and the cathodic delamination of PU coatings was significantly lessened as the MWCNT content increased to 0.5 wt%. Manas-Zloczower, et, al [14] performed the synthesis of PU nanocomposites via the in-situ polymerization of 1,4-Phenyldiisocyanate (PPDI) and Polycaprolactonediol in the addition of MWCNTs to investigate the effect of nanofillers on the reaction and system structural evolution. Gao, et, al [15] conducted the preparation of a flexible conductive polymer nanofiber composite (FCPNC) with the addition of the carbon nanotube (CNT), the good electrical conductivity and interconnected porous structure of the FCPNC made it possible to be used as a chemical vapor sensor."
Point 3: Please provide information about the production of the MWCNTs, do they content metals? Which quantities, surface area, etc….
Response 3: Considering the reviewer's comments, we have provided information about the production of the MWCNTs in the revised manuscript as"The MWCNTs (FloTube 9000 series) were supplied by Beijing Tiannai Technology Co., Ltd., Beijing, China. The purity, average diameter, average length and tap density of MWCNTs were respectively 95%–97.5%, 10–15 nm, 10 μm and 0.03–0.15 g/cm3."
Point 4: Describe the nozzle of the spray gun. Do you appreciate agglomeration on the tip?
Response 4: We are very sorry for our negligence of the description of the nozzle of the spray gun. It has been improved in the revised manuscript as"When it comes to electrostatic spraying (ES), there is a nozzle with a spiculate edge in the head of the spray gun, where would instantly generate high-voltage discharge as well as air ionization once the high-voltage negative electricity is connected, then, the advantage of high-voltage corona discharge leads to the formation of electrostatic field between the spray gun and the metal substrate. Subsequently, the liquid coating with negative charges ejected from the nozzle would be attracted to the metal substrate with positive charges under the action of electrostatic attraction."
Point 5: The discussion of the dispersion of the MWCNTs by SEM is not sufficiently justified. Please provide information of aggregate sizes, etc… micrographs c and d are clearly inhomogeneous which may lead to not conclusive results.
Response 5: Thanks for the constructive and thoughtful tips. We have made correction according to the reviewer's comments in the revised manuscript as"The presence may be explained as follows. Thanks to the good atomization performance of ES the still agglomerated MWCNTs that had been treated by magnetic stirring and ultrasonic dispersion could be dispersed again. Thus, the agglomeration and sedimentation of MWCNTs could be weakened to some extent. Nevertheless, the difference should not be ignored if the coating was prepared by Br. When the MWCNT content was 0.3 wt% there were few MWCNTs in WPU resins in the top area of BrC (Figure 1C), resulting in BrC to be unable to conduct electricity. As the MWCNT content increased to 0.6 wt%, MWCNTs in BrC could contact with each other to form a valid conductive network (Figure 1D), however, there were apparently agglomerated and deposited MWCNTs due to their uneven distributions, thus, the structure of BrC would be less compact. Moreover, the dispersion of MWCNTs in WPU resins was further discussed by TEM. Figure 2A and B respectively show morphologies of MWCNTs in uncured ESC 0.6 wt% and BrC 0.6 wt% on the surfaces of the steel substrates. In comparison with the morphology of the MWCNT in ESC 0.6 wt% (Figure 2A) there were obvious agglomerated MWCNTs in BrC 0.6 wt% (Figure 2B) and the size of agglomerated MWCNTs particles was about 150-200 nm. The result was consistent with that of SEM."
| (in attachment) |
Figure 2. Morphologies of MWCNTs in uncured (A)ESC 0.6 wt% and (B)BrC 0.6 wt% on the steel substrates.
Point 6: Please provide the electrical conductivity of the coatings, no values are given in the manuscript which are necessary to discuss their properties.
Response 6: Thanks for the thoughtful and helpful comments. The values of the electrical conductivity of the coatings have been provided in Table 1 in the revised manuscript as follows:
Table 1. Electrical Conductivity of WPU coatings with different MWCNTs contents
Coatings | Properties | 0 wt% | 0.3 wt% | 0.6 wt% | |
ESC | Thickness(μm) | 81±4 | 82±4 | 83±4 | |
Square resistance(MΩ/□) | 0 | 156.2±5 | 2.6±0.2 | ||
Resistivity(Ω m) | 0 | 12808.4 | 215.8 | ||
BrC | Thickness(μm) | 82±4 | 82±4 | 83±4 | |
Square resistance(MΩ/□) | 0 | 0 | 155.7±0.5 | ||
Resistivity(Ω m) | 0 | 0 | 12923.1 | ||
Their properties have been discussed in the revised manuscript as"Table 1 summarizes the electrical conductivity of coatings. It was evident that ESC 0.3 wt% was completely capable to conduct electricity, but BrC 0.3 wt% failed to conduct electricity. The electrical conductivity of ESC was better than that of BrC. Generally, the result of the electrical conductivity of coatings was consistent with that of the dispersity of MWCNTs."
Special thanks to you for your good comments.

Reviewer 2 Report
The authors described multi-walled carbon nanotubes (MWCNTs)/waterborne polyurethane (WPU) abrasion-proof and conductive coatings prepared by electrostatic spraying. The work is suitable for theme of Special Issue "Carbon-Based Polymer Nanocomposites for High-Performance Applications" in Polymers. Nevertheless, I do have some concerns on the manuscript that I think would be helpful to any reader of the article if clarified.
In page line 57, the word "of" in the sentence "The dispersion of MWCNTs in the coating and the of electrical conductivity " should be deleted.
The spray rate and time should be provided in the experiment.
The accurate electrical conductivity should be provided in the Table 1.
In page 5 line 179-181, the authors claimed that "When the WPU coating was prepared by ES, the liquid coating was blown to the metal substrate through the compressed air and under the action of the impact force the coating structure would become denser after curing". What is the driving force of electrostatic spraying? Which is come from compressed air or electrostatic force? The authors should be clearly stated here.
In Figure 5, the authors only described Figure 5A-C. However, there is no any description and analysis on Figure 5D-F.
The language should be polished.
The applications of conductive multi-walled carbon nanotubes (MWCNTs)/waterborne polyurethane were not introduced clearly in this work. There some latest works focus on CNT based hydrogel and conducting polymer which are related to the present work, such as Advanced Science, 2017, 4, 1600190; NPG Asia Materials, 2017, 9, e437, recommended to the authors for discussion.
Author Response
Response to Reviewer 2 Comments
Dear editor and reviewer,
Thank you for your comments concerning our manuscript entitled " Mechanical properties of multi-walled carbon nanotubes/waterborne polyurethane conductive coatings prepared by electrostatic spraying " (Manuscript ID: polymers-481645). Thanks for the positive, helpful and constructive comments and suggestions. We elaborately revised the manuscript according to these comments, where the revised parts have been highlighted (in red) for the convenience of checking. The following responses are in accordance with the specific comments in the report.
If you have any questions regarding this revision, please contact me.
Sincerely yours,
Lajun Feng
A point-by-point reply to the reviewer's comments:
Point 1: In page line 57, the word "of" in the sentence "The dispersion of MWCNTs in the coating and the of electrical conductivity " should be deleted.
Response 1: Thanks for the suggestion. The word "of" in the sentence "The dispersion of MWCNTs in the coating and the of electrical conductivity " has been deleted and the sentence has been modified as"The dispersion of MWCNTs and the electrical conductivity, surface hardness and wear resistance of the coating were studied."in page 2 line 76-78.
Point 2: The spray rate and time should be provided in the experiment.
Response 2: Considering the reviewer's comment, the spray rate and time have been provided in the revised manuscript as"The feedwell diameter was 1 mm, the liquid flow rate was 2 ml/min and the spray time was 1–2 min."(see Section 2.2.1 for details)
Point 3: The accurate electrical conductivity should be provided in the Table 1.
Response 3: We are very sorry for our inaccurate description of the electrical conductivity. It has been provided in Table 1 in the revised manuscript as follows:
Table 1. Electrical Conductivity of WPU coatings with different MWCNTs contents
Coatings | Properties | 0 wt% | 0.3 wt% | 0.6 wt% | |
ESC | Thickness(μm) | 81±4 | 82±4 | 83±4 | |
Square resistance(MΩ/□) | 0 | 156.2±5 | 2.6±0.2 | ||
Resistivity(Ω m) | 0 | 12808.4 | 215.8 | ||
BrC | Thickness(μm) | 82±4 | 82±4 | 83±4 | |
Square resistance(MΩ/□) | 0 | 0 | 155.7±0.5 | ||
Resistivity(Ω m) | 0 | 0 | 12923.1 | ||
Their properties have been discussed in the revised manuscript as"Table 1 summarizes the electrical conductivity of coatings. It was evident that ESC 0.3 wt% was completely capable to conduct electricity, but BrC 0.3 wt% failed to conduct electricity. The electrical conductivity of ESC was better than that of BrC. Generally, the result of the electrical conductivity of coatings was consistent with that of the dispersity of MWCNTs."
Point 4: In page 5 line 179-181, the authors claimed that "When the WPU coating was prepared by ES, the liquid coating was blown to the metal substrate through the compressed air and under the action of the impact force the coating structure would become denser after curing". What is the driving force of electrostatic spraying? Which is come from compressed air or electrostatic force? The authors should be clearly stated here.
Response 4: Thanks for the constructive suggestion, we are very sorry for our unclear statement. The sentence "When the WPU coating was prepared by ES, the liquid coating was blown to the metal substrate through the compressed air and under the action of the impact force the coating structure would become denser after curing" in the original manuscript has been improved as"The reason may be explained by the followings. Generally, the reason of the structure of ESC being more compact might be that under the action of high-voltage electrostatic field the defects which were generated in the coating during the former spray could be filled up by the atomized coating during the latter spray. When compressed air was used as the driving force to transport the liquid coating, however, the liquid coating was attracted to the metal substrate not only by electrostatic attraction during spray, it could also be more effectively atomized by the impact force of compressed air to prevent conductive fillers from agglomeration. Thus, the denser coating structure was formed after curing."in the revised manuscript (see page 6 line 235-243 for details).
Furthermore, the working principle of ES has been improved in"Section 1. Introduction"in the revised manuscript as"When it comes to electrostatic spraying (ES), there is a nozzle with a spiculate edge in the head of the spray gun, where would instantly generate high-voltage discharge as well as air ionization once the high-voltage negative electricity is connected, then, the advantage of high-voltage corona discharge leads to the formation of electrostatic field between the spray gun and the metal substrate. Subsequently, the liquid coating with negative charges ejected from the nozzle would be attracted to the metal substrate with positive charges under the action of electrostatic attraction. When compressed air was used as the driving force to transport the liquid coating, the coating could be more effectively atomized with the impact force of compressed air during spraying to avoid the agglomeration of conductive fillers."
Point 5: In Figure 5, the authors only described Figure 5A-C. However, there is no any description and analysis on Figure 5D-F.
Response 5: Thanks for the thoughtful and constructive tips. "Figure 5"in the original manuscript has been modified as "Figure 6" and the description and analysis on Figure 5D-F have been added in the revised manuscript as"By the analysis of the wear rate and friction coefficient curves of BrC (Figure 4 and 5), it was concluded that the wear resistance of BrC was worse than that of ESC and the Friction coefficient-Time curve of BrC was greatly wavy. This may be due to the uneven dispersion of MWCNTs and the poor compactness of BrC. In addition, the upward trend of agglomerated MWCNTs with the increase of the MWCNT content may cause the growth of micro-defects of BrC. As the friction time increased there were worn pieces increasingly generating in the coating. It would come into the wear track to be a kind of abrasive materials which may increase the wear rate and friction coefficient of BrC. Figure 6D, E and F show wear track morphologies of BrC after wear test. It was evidently that the wear track of BrC was wider and deeper than that of ESC and the number of worn pieces on the wear track of BrC was more in comparison with that of ESC. Additionally, the wear degree of BrC was aggravated with a growth in the MWCNT content."(see page 9 line 350-361 for details)
Point 6: The language should be polished.
Response 6: Thanks for the kind and helpful suggestion. The authors have checked the manuscript carefully. In the revised manuscript, the English in the whole article has been modified and some grammatical mistakes have also been corrected.
Point 7: The applications of conductive multi-walled carbon nanotubes (MWCNTs)/waterborne polyurethane were not introduced clearly in this work. There some latest works focus on CNT based hydrogel and conducting polymer which are related to the present work, such as Advanced Science, 2017, 4, 1600190; NPG Asia Materials, 2017, 9, e437, recommended to the authors for discussion.
Response 7: Thanks for the constructive and helpful comments. We have made correction according to the reviewer's comments in the revised manuscript as"Furthermore, it would provide a theoretical basis for the preparation and application of MWCNTs/WPU abrasion-proof and conductive coatings.". In addition, the authors have added the related references [20] and [21] in the revised manuscript:
Cai, G.F.; Wang, J.X.; Lin, M.F.; Chen, J.W.; Cui, M.Q.; Qian, K.; Li, S.H.; Cui, P.; Lee, P.S. A semitransparent snake-like tactile and olfactory bionic sensor with reversibly stretchable properties. NPG Asia Mater. 2017, 9, e437.
Cai, G.F.; Wang, J.X.; Qian, K.; Chen, J.W.; Li, S.H.; Lee, P.S. Extremely stretchable strain sensors based on conductive self-healing dynamic cross-links hydrogels for human-motion detection. Adv. Sci. 2017, 4, 1600190.
Special thanks to you for your good comments.

Round 2
Reviewer 1 Report
Dear authors,
I feel satisfied with all the modifications included in the manuscript. Although the work could be further improved it offers valuable information for potential readers of Polymers. I do recommend it for publication and encourage the authors to continue exploring the us of CNTs in different substrates.
With kind regards